# Myocardial Strain during Surveillance Screening Is Associated with Future Cardiac Dysfunction among Survivors of Childhood, Adolescent and Young Adult-Onset Cancer

**DOI:** 10.3390/cancers15082349

**Published:** 2023-04-18

**Authors:** Wendy J. Bottinor, Xiaoyan Deng, Dipankar Bandyopadhyay, Gary Coburn, Corey Havens, Melissa Carr, Daniel Saurers, Chantelle Judkins, Wu Gong, Chang Yu, Debra L. Friedman, Scott C. Borinstein, Jonathan H. Soslow

**Affiliations:** 1Department of Internal Medicine, Division of Cardiovascular Medicine, Virginia Commonwealth University, Richmond, VA 23298, USA; 2Division of Cardiovascular Medicine, Department of Medicine, Vanderbilt University School of Medicine, Nashville, TN 37232, USA; 3Department of Biostatistics, Virginia Commonwealth University, Richmond, VA 23298, USA; 4Department of Pediatrics, Division of Pediatric Cardiology, Vanderbilt University, Nashville, TN 37232, USA; 5Department of Pediatrics, Division of Hematology-Oncology, Vanderbilt University Medical Center, Nashville, TN 37232, USA; 6Department of Biostatistics, Vanderbilt University, Nashville, TN 37232, USA

**Keywords:** childhood cancer survivors, myocardial strain, cardio-oncology, heart failure, anthracyclines

## Abstract

**Simple Summary:**

The optimal cardiovascular screening strategy among survivors of cancer diagnosed at age 39 years or younger has not been determined. This retrospective analysis examined the role of echocardiography-based myocardial strain in identifying survivors at risk for left ventricular dysfunction. In this population, longitudinal and circumferential strain can likely improve the identification of survivors at risk for cardiovascular dysfunction and provide an opportunity for early intervention.

**Abstract:**

Cardiovascular disease is a leading contributor to mortality among childhood, adolescent and young adult (C-AYA) cancer survivors. While serial cardiovascular screening is recommended in this population, optimal screening strategies, including the use of echocardiography-based myocardial strain, are not fully defined. Our objective was to determine the relationship between longitudinal and circumferential strain (LS, CS) and fractional shortening (FS) among survivors. This single-center cohort study retrospectively measured LS and CS among C-AYAs treated with anthracycline/anthracenedione chemotherapy. The trajectory of LS and CS values over time were examined among two groups of survivors: those who experienced a reduction of >5 fractional shortening (FS) units from pre-treatment to the most recent echocardiogram, and those who did not. Using mixed modeling, LS and CS were used to estimate FS longitudinally. A receiver operator characteristic curve was generated to determine the ability of our model to correctly predict an FS ≤ 27%. A total of 189 survivors with a median age of 14 years at diagnosis were included. Among the two survivor groups, the trajectory of LS and CS differed approximately five years from cancer diagnosis. A statistically significant inverse relationship was demonstrated between FS and LS −0.129, *p* = 0.039, as well as FS and CS −0.413, *p* < 0.001. The area under the curve for an FS ≤ 27% was 91%. Among C-AYAs, myocardial strain measurements may improve the identification of individuals with cardiotoxicity, thereby allowing earlier intervention.

## 1. Introduction

Cardiovascular disease is a leading contributor to late morbidity and mortality after cancer treatment [1,2,3]. When compared with age-matched peers, cardiovascular-related death is approximately seven-fold higher in childhood cancer survivors, and at least three-fold higher in adolescent and young adult survivors [2,3]. The use of serial cardiac imaging to monitor cardiac dysfunction in childhood, adolescent, and young adult survivors (C-AYAs) is supported by several professional societies; however, the optimal imaging strategy for cardiac surveillance is undetermined [4,5,6,7,8].

Quantitative two-dimensional (2D) assessments of left ventricular (LV) systolic function measured by left ventricular ejection fraction (LVEF) or fractional shortening (FS) have historically been the focus of cardiac surveillance. Unfortunately, the ability of LVEF and FS to identify asymptomatic survivors who are at risk for treatment-related cardiotoxicity can be limited [9,10,11]. Additionally, evidence suggests that medical therapy is most efficacious at earlier stages of disease, with a decline in efficacy occurring within months of delayed treatment [12]. Changes in LVEF and FS are often late findings of cardiotoxicity, detectable only after significant heart dysfunction has already occurred [13,14]. LVEF and FS, therefore, may not be optimal metrics for cardiac surveillance in C-AYAs.

LV strain, which is an assessment of myocardial deformation, can indicate cardiovascular dysfunction prior to a decline in LV systolic function. In patients with adult-onset malignancies, baseline longitudinal strain (LS) and early changes in LS during therapy are superior metrics to identify patients at a higher risk for subsequent cardiovascular events, including heart failure [9,14,15,16]. In C-AYAs, abnormalities in strain can be detected during and after the completion of cancer treatment [17,18]. The role of advanced imaging techniques, including LV strain, in the long-term cardiac surveillance of C-AYAs treated with cardiotoxic cancer therapy is yet to be determined and is an area of active investigation [19,20,21].

We hypothesized that in C-AYAs, LS and circumferential left ventricular systolic strain (CS) would be associated with future FS values. To test this hypothesis, we performed a retrospective analysis of LS and CS in a cohort of C-AYAs treated with cardiotoxic chemotherapy.

## 2. Materials and Methods

This retrospective cohort analysis was reviewed and approved by the Institutional Review Board. An existing database of 484 patients with sarcoma, Hodgkin lymphoma, or acute myeloid leukemia treated at Vanderbilt University Medical Center between 2001 and 2019 was reviewed. Subjects were included if they were treated by the pediatric hematology/oncology program with anthracyclines or anthracenediones, such as mitoxantrone, and had adequate clinical records to determine diagnosis, therapeutic regimen, and cardiovascular history. Subjects were required to have one pre-therapy and one or more post-therapy echocardiograms available with DICOM images of adequate quality for retrospective analysis. Subjects were not included if clinical remission could not be achieved. If relapse occurred less than 1 year after the original end of treatment, the end of therapy date was defined as the date after completion of both initial and salvage therapy. If relapse occurred more than 1 year after the original end of treatment, the end of therapy date was defined as the date of completion of the initial treatment regimen. In this scenario, echocardiograms, cancer therapy, and cardiovascular outcomes prior to the date of relapse were included in the analysis.

Individuals for whom cardiac screening and management likely differed from the broader survivor population were excluded. This included individuals with pre-treatment echocardiograms demonstrating greater than mild mitral or aortic valve regurgitation/stenosis, greater than moderate tricuspid or pulmonary valve regurgitation/stenosis, hemodynamically significant congenital heart disease (not including PFO/small ASD or small VSD), greater than moderate pericardial effusion, constrictive physiology, an FS < 28% (as quantified by 2D or M Mode methods), an LVEF < 55% (as quantified by 5/6 area-length, biplane, or 3D methods), an arrhythmia (not isolated PACs or PVCs), a family history of cardiomyopathy, or evidence of non-compaction/infiltrative cardiomyopathy.

Data including gender, race, type of malignancy, age at diagnosis, total anthracycline/anthracenedione dose, and radiation to a field with potential impact on the heart were obtained from a review of the electronic health record. Doxorubicin isotoxic equivalents were calculated and relevant radiation fields were identified based on the Children’s Oncology Group Long-Term Follow-Up Guidelines Version 5 [4].

Echocardiograms performed at our institution from the time of cancer diagnosis and onwards were identified. Echocardiograms were performed on commercially available equipment (EPIQ7 and iE33 [Philips Medical Systems, Andover, MA, USA], ACUSON Sequoia [Siemens Medical Solutions, Malvern, PA, USA]). Clinical echocardiogram reports were reviewed. The pertinent clinical findings, including measurements of LVEF and FS, and evidence of valvular disease, pericardial disease, or other cardiac pathology were recorded.

DICOM images were imported into a research database and analyzed using Cardiac Performance Analysis 4.6 (TomTec Imaging Systems, Unerschleissheim, Germany). For all studies with adequate image quality, LS and CS were measured by tracing endocardial borders in the apical four-chamber view and short-axis view at the mid-papillary level. Image analysis was performed and reviewed by a core group of 3 sonographers and 1 cardiologist who were blinded to the clinical outcome. All data were compiled and stored in REDCap, a HIPAA compliant electronic database [22].

### Statistical Analysis

To determine the trajectory of LS and CS over time, survivors were divided into two groups. Based on reproducibility data indicating a decline in FS by >5 absolute units is clinically significant [23], we dichotomized our FS data into two groups according to the presence or absence of a drop in FS > 5 units on the most recent echocardiogram in comparison to the pre-treatment echocardiogram. We then examined the relationship between this dichotomized variable, LS, and CS over time.

The random intercept mixed model was used in this longitudinal analysis to determine the relationship between CS, LS and FS as time proceeds. All available time points were included in the analysis. In addition to LS and CS, the following demographic and treatment factors were identified a priori and included as explanatory variables in the analyses: age at diagnosis, total doxorubicin isotoxic equivalents, and radiation to a field involving the heart (yes/no). Given a variable duration of follow-up, the time elapsed since diagnosis was also included in the model. The two-sided significance level was set to 5% for assessing the significance of the estimated parameters.

We assessed the relationship between FS and strain measurements using three models: (a) LS individually; (b) CS individually; and (c) LS and CS combined. The root mean square error (RMSE) was determined for all three models and the Diebold–Marino test was used to compare the predictive accuracy of the 3 models [24].

We performed a post hoc analysis using a receiver operator characteristic curve to determine the ability of our model to correctly predict an FS ≤ 27%, a clinically accepted marker ofabnormal left ventricular systolic function [25]. The analyses were conducted using SAS (Statistical Analysis System, version 9.4) software. The model cross-validation and comparisons were performed with R (version 4.0.5).

## 3. Results

Our initial chart review identified 206 unique subjects who met inclusion criteria. Among subjects who were not included, the most common reasons were unavailable digital echocardiograms (72%) and not achieving remission (17%). The remaining reasons for exclusion were unclear treatment history, no exposure to anthracyclines, and significant congenital heart disease.

A total of 199 subjects had DICOM images of adequate quality with appropriate tracking of LV motion to perform measurements of left ventricular strain (Figure 1). Due to the retrospective nature of this study, 189 subjects had serial FS values, while only 83 had serial LVEF values. Therefore, we chose to focus our analysis on the individuals with serial FS values.

### 3.1. Demographic and Clinical Characteristics

Among these individuals, 55% were male. The median age at diagnosis was 14 years, and the median current age was 20 years. The most common cancer diagnoses were Hodgkin lymphoma 30% (56/189), acute myeloid leukemia 24% (45/189), and Ewing sarcoma 22% (42/189). The median doxorubicin isotoxic equivalence dose was 300 mg/m^2^ IQR (200 mg/m^2^ to 375 mg/m^2^). A total of 32.3% (61/189) of the participants had received radiation. The median follow-up time was 1.8 years with an IQR of 0.8 to 4.8 years (Table 1). Approximately 40% of the population was followed for at least 3 years and 44 participants (23%) had follow-up echocardiograms ≥ 5 years from diagnosis.

### 3.2. Left Ventricular Function

The pre-treatment median FS was 37.1% IQR (34.7% to 40.1%). The pre-treatment median LS was −19.8% IQR (−21.9% to −18.1%) with a pre-treatment median CS of −25.6% IQR (−28.6% to −22.9%). The median FS on the most recent follow-up was 35.0% IQR (32.0% to 38.0%) with a median LS of −18.7% IQR (−20.9% to −16.7%) and a median CS of −24.3% IQR (−26.9% to −21.9%) (Table 1).

### 3.3. Longitudinal FS Values

By plotting LS over time for both FS groups, we determined that approximately five years from cancer diagnosis, the rate of change in LS began to differ between the two groups with a larger rate of decline among individuals with a reduction of >5 FS units when compared with individuals without a reduction of ≥5 FS units (Figure 2a). Similarly, the rate of change in CS began to differ between the two groups after approximately five years (Figure 2b). Despite these changes in LS and CS, the median FS was similar between the two groups at five years (34.3% among those with an eventual decline of >5 FS units versus 34.6% among those with an eventual decline of ≤5 FS units).

### 3.4. Mixed-Effects Modeling

Using a mixed-effects model, we identified significant inverse relationships between FS and the following variables: time since diagnosis, LS, and CS (Table 2). For every year since diagnosis, FS declined by 0.36 units (*p* < 0.001). For every one-unit increase in LS and CS, FS decreased by 0.13 and 0.41 units, respectively. We found no significant interaction between time and LS or CS. Figure 3 depicts the estimated FS based on our model.

Next, we examined LS and CS individually. By comparing plots of LS and estimated FS to plots of CS and estimated FS, we determined that there was a higher rate of change between CS and estimated FS when compared to LS and estimated FS (Figure 4a,b). This led us to further examine LS and CS.

### 3.5. Cross Validation

To better understand the relationship between LS, CS, and FS, we determined the RMSE for all three models based on LS alone (RMSE 4.21), CS alone (RMSE 4.07), and both LS and CS (RMSE 4.05). Only the model that included both LS and CS did not violate the normality assumptions. We also assessed differences in the forecasted accuracy of the three models using the Diebold–Mariano test (Table 3). The Diebold–Mariano test indicated no significant difference in the forecasted accuracy of the three models. Therefore, we selected a model that included LS and CS for further assessment with a receiver operator characteristic curve.

### 3.6. Prediction of LV Dysfunction

Our post hoc analysis used an FS value of ≤27% as a marker of clinically significant LV dysfunction. The area under the curve (AUC) for the receiver operating characteristic curve (ROC) generated by our data is 0.906 (95% confidence interval: 0.850–0.962) with an AUC of 0.900 for the corresponding smoothed ROC (95% confidence interval: 0.816–0.946). (Figure 5). This implies that the model is adequate in distinguishing an FS value of ≤27% compared to ≥27%.

## 4. Discussion

The role of strain in cardiovascular screening among C-AYAs is poorly defined. Our study specifically examines the association between LS, CS, and FS in a retrospective single-center cohort. Our main finding shows that a model using LS and CS to estimate future FS can accurately identify survivors at risk for LV dysfunction (defined as an FS ≤ 27%). Additionally, the rate of change in LS and CS starting five years post-cancer diagnosis may differ between survivors who develop a clinically significant decline in FS and those who do not. Therefore, using LS and CS during cardiac screening may identify survivors at a higher risk of future cardiotoxicity, and thereby provide an opportunity for earlier clinical intervention.

Although FS and LVEF are commonly used during cardiac screening, evidence suggests that relying on these late biomarkers of cardiotoxicity is likely suboptimal. Cardinale et al. has demonstrated that 36% of adults treated with anthracycline-based chemotherapy who develop an LVEF ≤ 45% will not achieve complete recovery of LV systolic function, even if heart failure therapy begins immediately [12]. Similarly, in a population of childhood survivors with significant LV systolic dysfunction prior to starting heart failure therapy, a transient improvement in cardiac function was observed; however, this benefit was not sustained through the 10 years of follow-up [26]. Our study shows that myocardial strain is a potential earlier predictor of cardiotoxicity among C-AYAs undergoing cardiac screening and consideration should be given to adding strain to the current imaging protocols.

Strain can also provide a more effective strategy for timing the initiation of medical therapy. In an adult population, an LS-guided strategy for initiation of heart failure therapy (≥12% relative reduction in LS) during cancer treatment has demonstrated superior preservation of cardiac function compared to an LVEF-guided strategy (>10% absolute reduction in LVEF) [16]. Promising results from a retrospective single-center study of 22 childhood survivors using a strain-guided approach for early treatment of cardiotoxicity demonstrated sustained improvement in strain values among survivors treated with ACEI or angiotensin receptor blocker (ARB) therapy [27]. Here, our work also suggests that implementing strain measurements in cardiac screening for C-AYAs could provide an opportunity for a timelier intervention.

There are population-specific considerations when utilizing strain measurements. LS is preferentially emphasized over CS in adult-based cardio-oncology guidelines. However, studies have indicated that CS values were abnormal more often than LS values in C-AYAs greater than one year after treatment [17,28]. In this study, we have demonstrated that a model including both LS and CS is optimal for C-AYAs.

Another unique aspect of cardiovascular screening in C-AYAs is the longevity of survivors, leading to an extended duration for follow-up and serial surveillance. Based on our data, changes in the trajectory of myocardial strain may begin to differentiate between lower- and higher-risk groups approximately five years from a cancer diagnosis, making this an ideal time to enhance screening.

### Limitations

Our study suggests a potential role for strain as an imaging biomarker for identifying survivors at a higher risk for the development of subsequent LV dysfunction. However, several limitations should be noted when interpreting these results: (a) The largest limitation is the retrospective nature of this study, which caused variability in available retrospective echocardiogram images and image quality. Although this limitation is not unique to our study and is common to retrospective imaging studies in this population, this hindered our ability to perform a Simpsons’ bi-plane measurement of ejection fraction [29]. Future studies utilizing cardiac magnetic resonance imaging should be considered to address this limitation. (b) Information on dexrazoxane use was not uniformly available and, thus, not accounted for in this study. This is significant given the potential cardio-protective benefits of this agent [30,31]. (c) Obesity can influence strain, but it was not accounted for in our model. However, only 15% of our population was obese, suggesting that this factor may be less influential in this cohort. (d) Finally, the generalizability of our results may be limited since this was a single-center study.

## 5. Conclusions

In a retrospective single-center study of C-AYAs, we showed that measurements of LS and CS could provide estimates of future FS. The change in the trajectory of LS and CS values among patients who developed cardiotoxicity became apparent approximately five years after a cancer diagnosis. These results will require further validation in a large prospective cohort but suggest the potential utility of strain to stratify the risk for LV dysfunction among C-AYAs undergoing cardiac surveillance.

## Figures and Tables

**Figure 1 cancers-15-02349-f001:**
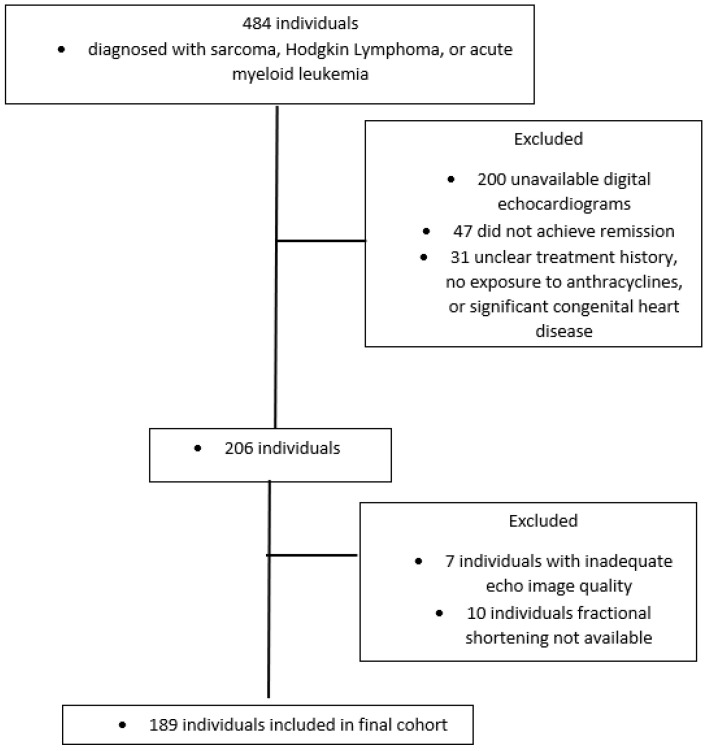
A total of 189 subjects were included in the study. The most common reasons for exclusion were unavailable digital echocardiograms (72%) and remission was not achieved (17%).

**Figure 2 cancers-15-02349-f002:**
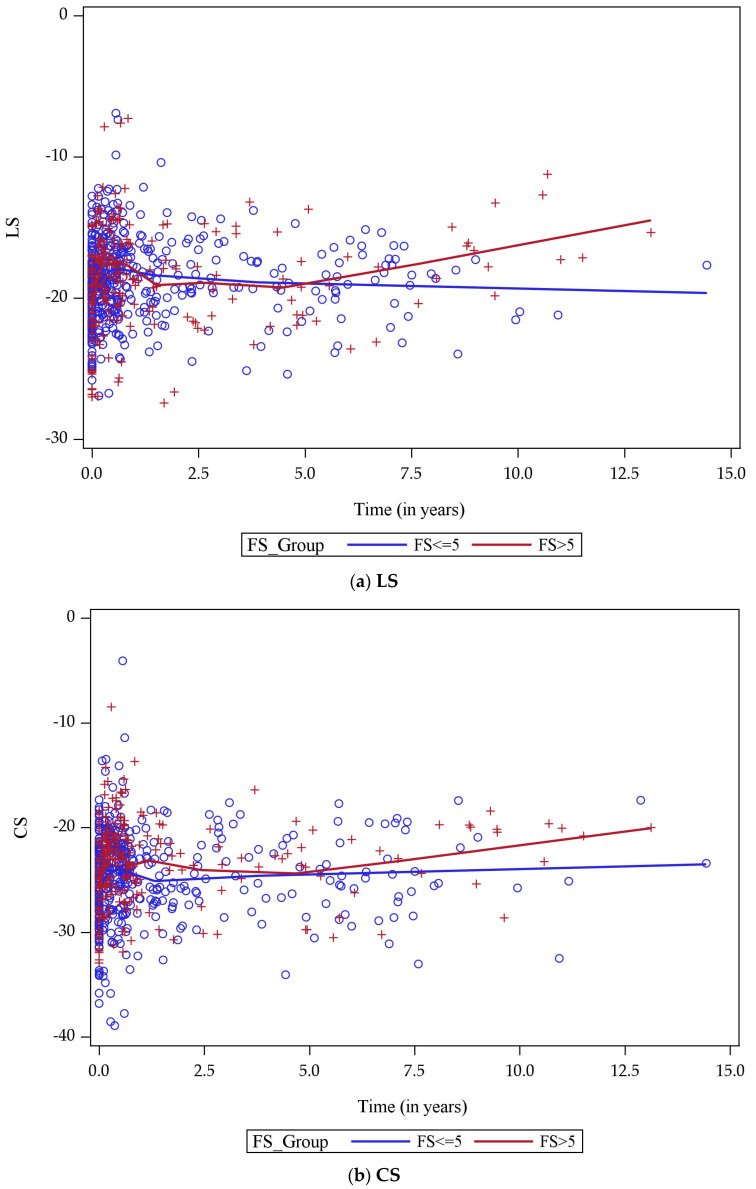
(**a**,**b**) Trends in longitudinal and circumferential strain over time. Longitudinal strain (LS) (**a**) and Circumferential strain (CS) (**b**) plotted over time among individuals who experienced a decline in fractional shortening (FS) from pre-treatment to the most recent echocardiogram by >5 FS unit (red) and those who did not (blue) indicates a difference in rate of change in LS and CS about five years after a cancer diagnosis. Separate Loess smoothers are applied to each group to visually demonstrate the longitudinal trends. +/◦ Indicate individual strain measurements.

**Figure 3 cancers-15-02349-f003:**
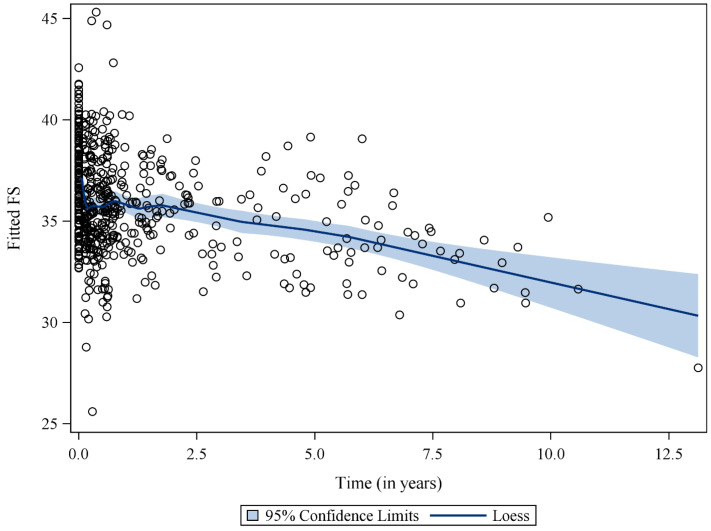
Estimated Fractional Shortening Using a Mixed-Effects Model. Fitted fractional shortening (FS) values (circles) based on a mixed-effects model, including longitudinal strain (LS), circumferential strain (CS), age at diagnosis, time since diagnosis, total doxorubicin isotoxic equivalents, and radiation to a field involving the heart (yes/no) across ‘time since diagnosis’. Loess-smoothed curve of the fitted FS values, along with the corresponding 95% confidence band, is also provided.

**Figure 4 cancers-15-02349-f004:**
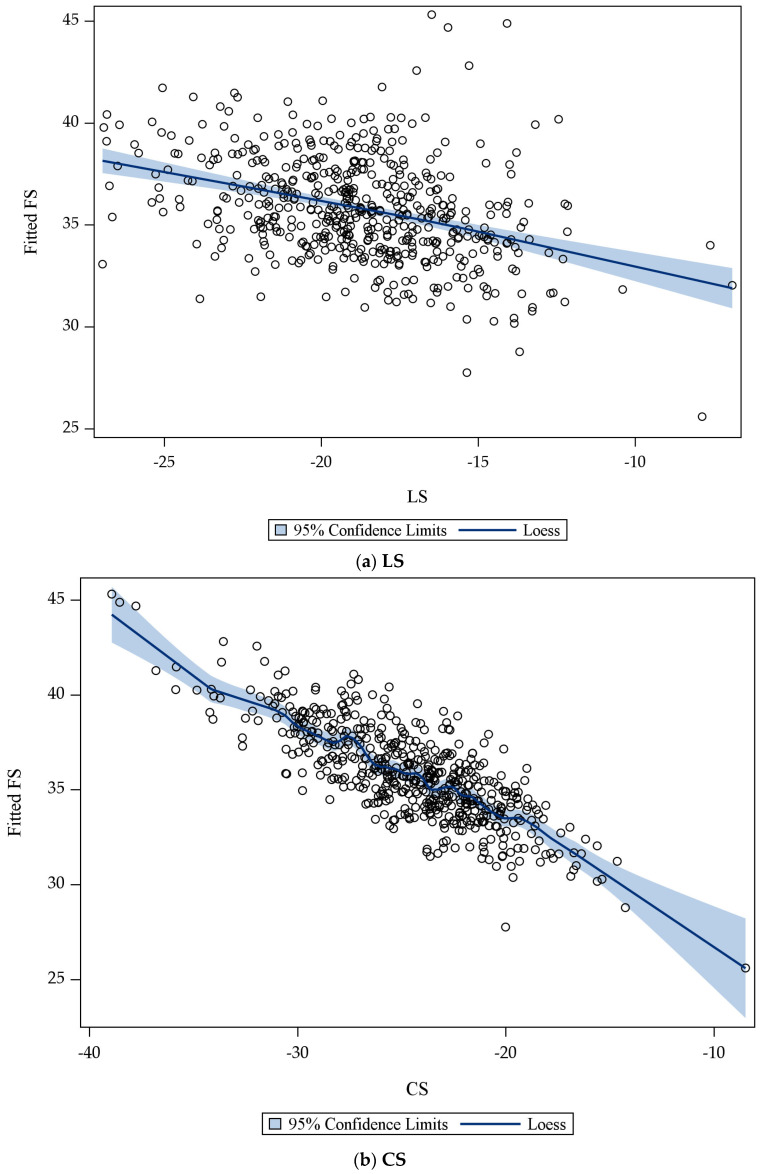
(**a**,**b**). Estimated Fractional Shortening Using Longitudinal or Circumferential Strain. Fitted fractional shortening (FS) values (circles) across longitudinal strain (LS) values (**a**) and circumferential strain (CS) values (**b**). Loess-smoothed curve of the fitted FS values, along with the corresponding 95% confidence band, is also provided.

**Figure 5 cancers-15-02349-f005:**
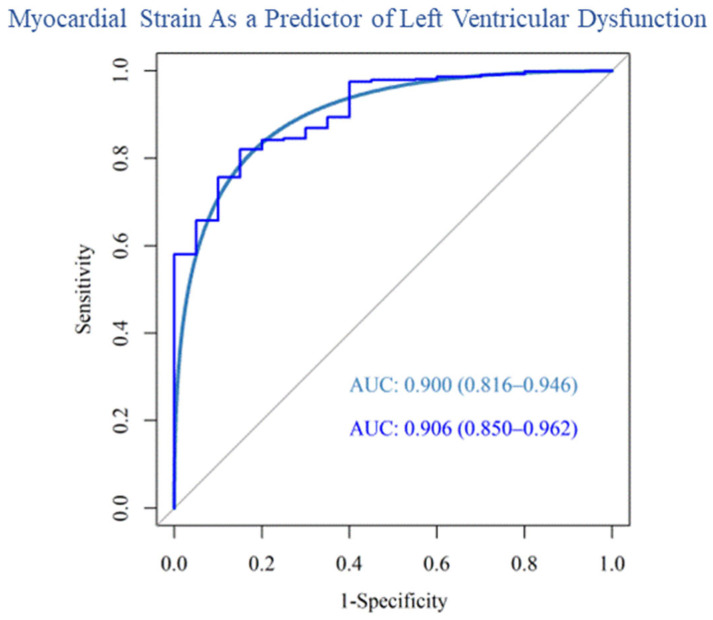
Predictive Ability of a Mixed-Effects Model to Identify Survivors at Risk for Left Ventricular Dysfunction. Receiver operator characteristics curve with smoothing (light blue) and without smoothing (dark blue) is shown. The area under the curve (AUC) for the receiver operator characteristic curve (ROC) generated by our data is 0.906 (95% confidence interval: 0.850–0.962) with an AUC of 0.900 for the corresponding smoothed ROC (95% confidence interval: 0.816–0.946).

**Table 1 cancers-15-02349-t001:** Demographic and Treatment Characteristics of Survivors.

Characteristics	N = 189
Male (%)	104 (55.0%)
Age at diagnosis (Year, Median)	14.4
Age at Last Echocardiogram (Year, Median)	16.7
Cancer Type	
Acute myeloid leukemia	45 (23.8%)
Ewing sarcoma	42 (22.2%)
Hodgkin lymphoma	56 (29.6%)
Osteosarcoma	35 (18.5%)
Soft tissue sarcoma	11 (5.8%)
Cumulative Anthracycline Dose [mg/m^2^ Median (IQR)]	300 (200–375)
Radiation (%)	61 (32.3%)
Pre-treatment FS [Median (IQR)]	37.1% (34.7% to 40.1%)
Pre-treatment LS [Median (IQR)]	−19.8% (−21.9% to −18.1%)
Pre-treatment CS [Median (IQR)]	−25.6% (−28.6% to −22.9%)
Most recent FS [Median (IQR)]	35.0% (32.0% to 38.0%)
Most recent FS < 27%	10 (5.3%)
Most recent LS [Median (IQR)]	−18.7% (−20.9% to −16.7%)
Most recent CS [Median (IQR)]	−24.3% (−26.9% to −21.9%)
Follow-up time (Years) [Median (IQR)]	1.8 (0.8 to 4.8)

**Table 2 cancers-15-02349-t002:** Mixed-Effects Model Including All Variables Identified A Priori.

Predictor	Estimate	95% C.I.		*p*-Value
		Lower	Upper	
Age at diagnosis	−0.059	−0.1425	0.02548	0.1711
Time since diagnosis (years)	−0.362	−0.524	−0.2007	<0.0001
Doxorubicin isotoxic equivalence total dose	0.001	−0.0007	0.00203	0.3245
Chest Radiation (No)	0.136	−0.8238	1.0956	0.7803
LS	−0.129	−0.251	−0.00665	0.0388
CS	−0.413	−0.5088	−0.3179	<0.0001

**Table 3 cancers-15-02349-t003:** Root Mean Square Error (RMSE) and Forecasts Comparison Among Models with LS alone, CS alone, and LS with CS.

Model	RMSE	Model Forecasts Comparison	
LS alone	4.21	LS alone vs. CS alone	0.55
CS alone	4.07	LS alone vs. LS and CS	0.29
LS and CS	4.05	CS alone vs. LS and CS	0.43

## Data Availability

De-identified echocardiography measurements can be provided upon written request and IRB approval.

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
