# Peer review of "Myocardial Strain during Surveillance Screening Is Associated with Future Cardiac Dysfunction among Survivors of Childhood, Adolescent and Young Adult-Onset Cancer"

_cancers, 2023, doi:10.3390/cancers15082349_

Round 1

Reviewer 1 Report

Bottinor et al. evaluate the role or longitudinal and circumferential strain in predicting decreases in fractional shortening among survivors of childhood, adolescent and young-adult onset cancer treated with anthracyclines or anthracenediones. The study is timely and clinically relevant. The manuscript is well written.

Please consider the following suggestions in order to further improve the quality of the study:

1. 199 subjects had DIMOM images of adequate quality to calculate LV strain. At the same time it was reported that only 83 patients had LVEF values and that's why LVEF was not included in the analysis. Could you explain why LVEF could not be measured using the DICOM images? FS has several limitations in assessing LV systolic function related to loading conditions, wall motion abnormalities etc. LVEF is a widely accepted and routinely used measure of systolic dysfunction. 

2. Was radial strain measured? Was there a reason that in addition to GLS, only circumferential strain was measured?

3. The median follow up was 1.8 years. LS and CS changes became significant 5 years after cancer diagnosis. How many patients had follow up echos >=5 years from diagnosis? Have you shown that LS and CS precede the drop in FS? Or did they happen at the same time? If they preceded the drop in FS, by how long?

Reviewer 2 Report

This single center and retrospective study examined the clinical value of echocardiographic strain measurements in young cancer survivors (children, adolescents and young adults) who received anthracycline chemotherapy. Based on studies / research that shows that a reduction in ejection fraction and fractional shortening (FS) is commonly a relatively late finding of ventricular dysfunction, the authors used speckle tracking echocardiography to assess left ventricular strain.

189 patients were included and were divided into two groups. Patients with a drop in FS of more than 5 units on the latest echocardiogram compared to the pre-chemotherapy echocardiogram were compared with those who did not show that drop.

The authors found a greater decline in longitudinal and circumferential strain in patients with a drop in FS of more than 5 units about 5 years after cancer diagnosis.

There were also significant relationships between FS and time since diagnosis as well as FS and circumferentional strain (less for longitudinal strain).

In summary, these results support the use of echocardiographic strain measurements in cancer survivors during their long-term follow-up. Furthermore, it shows that paediatric cancer survivors should receive long-term cardiology follow-ups.

Methods:

1.      Did you re-measure FS or did you document those values that were measured at the time of echocardiographic examination?

2.      I understand that global values for LS and CS were measured. Did you consider also assessinig regional strain values?

Results:

3.      Was the image quality good enough to get results for both LS and CS in all 189 patients?

Discussion:

4.      I would welcome some preliminary recommendations. How often should strain measurements be performed etc.

5.      What about those with poor echo windows? Would you recommend other imaging methods?

Reviewer 3 Report

Please specify the following information:

1)     Materials and Methods An existing database of 484 patients with sarcoma, Hodgkin lymphoma, or acute myeloid leukemia treated at Vanderbilt University Medical Center between 2001 and 2019 was reviewed.

Median follow-up time was 1.8 years with an IQR of 0.8 to 4.8 years (Table 1).  Approximately 40% of the population was followed for at least 3 years. 

But in Figure 2a and 2b follow-up up to 15 years?

2). Receiver operator characteristics curve (Figure without a number) – what about Specificity and Sensitivity?

3). Radial strain – are you measure Radial strain? If no – Why no?

4). Limitations 

Our study suggests a potential role for strain as a biomarker – indicator?

Usually the word biomarket refers to biological indicators (blood, urine, etc.)
